# Effort–Reward Imbalance among a Sample of Formal US Solid Waste Workers

**DOI:** 10.3390/ijerph19116791

**Published:** 2022-06-01

**Authors:** Aurora B. Le, Abas Shkembi, Anna C. Sturgis, Anupon Tadee, Shawn G. Gibbs, Richard L. Neitzel

**Affiliations:** 1Department of Environmental Health Sciences, School of Public Health, University of Michigan, Ann Arbor, MI 48109, USA; ashkembi@umich.edu (A.S.); asturgis@umich.edu (A.C.S.); anupont@umich.edu (A.T.); rneitzel@umich.edu (R.L.N.); 2Department of Environmental and Occupational Health, School of Public Health, Texas A&M University, College Station, TX 77843, USA; sgibbs@tamu.edu

**Keywords:** effort–reward imbalance, psychosocial factors of work, waste workers, work stress

## Abstract

Background: Solid waste workers are exposed to a plethora of occupational hazards and may also experience work-related stress. Our study had three specific hypotheses: (1) waste workers experience effort–reward imbalance (ERI) with high self-reported effort but low reward, (2) unionized workers experience greater ERI, and (3) workers with higher income have lower ERI. Methods: Waste workers from three solid waste sites in Michigan participated in this cross-sectional study. We characterized perceived work stress using the short-version ERI questionnaire. Descriptive statistics and linear tests for trend were assessed for each scale. Linear regression models were constructed to examine the relationship between structural factors of work stress and ERI. Gradient-boosted regression trees evaluated which factors of effort or reward best characterize workers’ stress. Results: Among 68 participants, 37% of workers reported high effort and low reward from work (ERI > 1). Constant pressure due to heavy workload was most indicative of ERI among the solid waste workers. Union workers experienced 79% times higher ERI than non-unionized workers, while no significant differences were observed by income, after adjusting for confounders. Conclusions: Organizational-level interventions, such as changes related to workload, consideration of fair compensation, and increased support from supervisors, can decrease work stress.

## 1. Introduction

Waste workers are exposed to a plethora of occupational hazards but remain an overlooked worker population. Workplace exposures experienced by waste workers include physical (e.g., heat and cold stress, musculoskeletal disorders), chemical (e.g., heavy metals, methane, volatile organic compounds), and biological (e.g., human and animal bodily fluids and excreta, fungi) exposures [1,2,3]. In the US, there are approximately 460,000 formal workers who are part of the “Waste Management and Remediation Services” industry, as classified by the Bureau of Labor Statistics. Subsectors within this industry are establishments that provide local hauling of waste materials, manage healthcare waste, operate materials recovery facilities (i.e., sort recyclables or e-waste from the trash stream), provide remediation services (e.g., cleanup to contaminated sites), conduct septic pumping, and provide other miscellaneous waste management services. Ultimately, the responsibilities of workers fall under waste collection, waste treatment and disposal, or remediation and other services; related worksites range from small-scale formal waste collection to large industrial waste management services [4]. Waste workers in solid waste—waste generated from non-hazardous industrial, commercial, and residential activities—comprise the majority of this worker population in the US. In 2021, the number of total recordable injury and illness cases per 100 full-time workers in the waste industry was 3.5 compared to 2.7 per 100 full-time workers across all private industry employers, underscoring the occupational hazards faced by waste workers [4,5].

It is well-established that occupational hazards and psychosocial factors related to interactions between workers and their workplace environment influence physical and mental health [6,7]. Psychosocial factors of work include but are not limited to perceived work stress, burnout, work–family conflict, social support, decision latitude, job satisfaction, job security, and team climate [6,7,8,9]. For example, in a sample of critical care nurses, Rahman and colleagues found that stress, work–family conflict, and reward were good predictors of work-related fatigue and that work fatigue directly influenced injury, illness, and medical errors [9]. This demonstrates that psychosocial factors are indeed related to and can exacerbate occupational exposures.

Given that waste workers encounter numerous hazards in the workplace, we expect that they also experience work-related stress. Studies that have looked at psychosocial factors of work among waste workers found the prevalence of perceived stress among e-waste and solid waste workers to be high due to the nature of their work (e.g., ergonomic challenges, exposure to chemicals, limited personal protective equipment use). However, these studies were focused on West Africa and Southeast Asia [10,11,12]. To our knowledge, there have been no studies focused on psychosocial factors among United States solid waste workers, or among US waste workers overall.

In this study, we characterized perceived work stress among a sample of US waste workers using the effort–reward-imbalance (ERI) questionnaire [13]. In the ERI work model, chronic work-related stress results from the imbalance or non-reciprocity between high effort expended and low reward received. In other words, work stress arises when there is a perceived imbalance between high effort and low reward. Effort pertains to a worker’s job demands and obligations; reward can comprise both tangible and intangible items such as money, esteem, job security, and career opportunities [14,15]. The validated ERI questionnaire has been used to measure perceived work stress in a variety of populations (e.g., nurses, bus drivers, hotel housekeepers) to determine the contribution of structural and organizational contributors of work-related stress, rather than individual-level factors (e.g., sex, race/ethnicity) [16,17,18,19,20,21,22]. For our study, we had three specific hypotheses: (1) waste workers experience effort–reward imbalance with high self-reported effort but low reward, (2) workers who are part of a union experience greater effort–reward imbalance, and (3) workers with higher income have lower effort–reward imbalance.

## 2. Materials and Methods

### 2.1. Study Participants

Adult waste workers, aged 18 and above, from three solid waste sites in Michigan were recruited in this cross-sectional study in the fall of 2021 during the COVID-19 pandemic through convenience and referral sampling. Research staff contacted and recruited site supervisors of 39 solid waste sites throughout southeast Michigan using publicly available contact information. Site supervisors who consented to participate assisted in recruitment of their workers. Participants came from three diverse solid waste sites: (1) a local-level, small-scale, family-owned waste disposal site providing hauling services where all workers onsite participated in the study; (2) a county-level, recycling-only, waste management authority where all workers onsite participated in the study; and (3) a large-scale industrial waste management company with both hauling and landfill divisions where about half of employees onsite participated in the study. While only the large-scale industrial waste management company within the landfill division had unionized workers, not all landfill division workers were unionized. Each participant was asked to provide information on occupational biohazard exposures and levels of work stress. The extent to which the waste industry could benefit from incorporating the National Institute for Occupational Safety and Health (NIOSH) Total Worker Health (TWH) framework was also assessed. This paper focuses specifically on the levels of work stress reported by participants. Participants received $40 as compensation. All study procedures were approved by the University of Michigan Institutional Review Board (#HUM00202683).

### 2.2. Survey Questionnaire

Surveys were filled out either at the beginning or end of workers’ work shifts and took approximately 20 minutes to complete 74 items. Demographics such as age, income, work experience, and sex were collected. A part of the survey was dedicated to measuring work stress through the short-version effort–reward imbalance (ERI) questionnaire [13]. The ERI short-version questionnaire, developed by Siegrist, Li, and Montano, has been operationalized to determine workers’ self-reported effort, reward, and over-commitment in various occupational settings [13]. This questionnaire consists of three scales validated for their psychometric properties (e.g., scale reliability, factorial structure, convergent validity, discriminant validity, criterion validity, sensitivity to change over time) and includes: (1) a 3-item effort scale, (2) a 7-item reward scale, and (3) a 6-item over-commitment scale. Each item is presented on a 4-point scale from “strongly disagree” to “strongly agree”.

### 2.3. Survey Items

Effort was measured by assessing whether workers felt constant time pressure due to heavy workload (ERI1), had many interruptions/disturbances while performing their job (ERI2), and if their job had become more demanding over the past few years (ERI3). Each item was assigned a numeric value from 1 (“strongly disagree”) to 4 (“strongly agree”) and summed to construct a participant-level effort score.

Four items from the reward scale were positively phrased, such as “I receive the respect I deserve from my superior or a respective relevant person” and “Considering all my efforts and achievements, my job promotion prospects are adequate.” The remaining two items were negatively phrased, such as “My job promotion prospects are poor”, and “I have experienced or I expect to experience an undesirable change in my work situation.” Positively phrased items were assigned a numeric value from 1 (“strongly disagree”) to 4 (“strongly agree”), while negatively phrased items were reverse-scored. The sum of each item was taken to construct a participant-level reward scale.

Effort and reward were examined in contrast to one another to construct an effort–reward imbalance score using Equation (1), as defined by Siegrist, Li, and Montano:(1)$$\mathrm{ERI}=\frac{E}{R\times c}$$
where *E* = effort score, *R* = reward score, and *c* = 5/11 as a correction factor for the unequal number of items between the effort and reward scales. An ERI score = 1 signifies the same effort for the same reward; ERI > 1 indicates more effort for each reward; and ERI < 1 indicates less effort for each reward. As previously mentioned, the ERI score has been used as a proxy for work stress [18,19,20].

Negatively phrased over-commitment items included, “As soon as I get up in the morning I start thinking about work problems” and “Work rarely lets me go; it is still on my mind when I go to bed”, while one item, “When I get home, I can easily relax and ‘switch off’ work” was positively phrased. Negatively phrased items were assigned a value from 1 (“strongly disagree”) to 4 (“strongly agree”), while the positively phrased item was reverse-scored. All items were then summed by participants to construct an over-commitment score.

### 2.4. Statistical Analyses

All data cleaning and statistical analyses were conducted in RStudio using R v3.6.3 (Boston, MA, USA). Cronbach’s alpha coefficients were calculated for the three scales to assess internal reliability consistency. Item-total correlation coefficients were calculated to examine the consistency of items defining their respective scale. Descriptive statistics were computed for scores of each psychometric scale, along with exceedance fractions. Linear tests for trend were assessed for each scale when examining differences between groups through linear regression models and treating the groups as numerical variables (threshold of significance, two-sided *p* < 0.05). Bivariate and multivariate linear regression models were constructed to examine the relationship between structural factors of work stress (unionization, job site, and income) and ERI. Multivariate models were all adjusted for work experience, education, and sex, all of which may impact work stress. ERI scores were right-skewed (using a Shapiro–Wilk normality test, *p* < 0.05); as such, hypothesis testing and regression models utilized log-transformed ERI scores to validate the assumptions of linear regression. For this reason, relative risks (RRs) were assessed.

Lastly, gradient-boosted regression trees (BRTs) were incorporated to evaluate which factors of effort or reward best characterize the stress waste workers experience [23]. BRTs are a type of supervised machine learning that grows many regression trees in an iterative manner. Each tree uses recursive binary splits in a top-down, greedy approach to select a partition of a predictor that minimizes the error of the model. The depth of the tree can be specified to indicate the degree of interaction between each predictor. In this case, an interaction depth of two was specified to allow for two-way interactions between each predictor. In this analysis, this process was repeated 500 times, such that 500 regression trees were grown. However, no regression tree was run on information from all data points. Rather, a random subset of 70% of data points was provided for each regression tree. As such, the predictors chosen for each binary split are not always the same, as they may not be the optimal partition for a given subset of the data. This helps to overcome the shortcomings of a small sample size by simulating 500 random subsets of the study population, which may mimic other workplace scenarios.

The number of times a predictor was chosen over the 500 trees and the amount of error it removed from the model (weighted to the number of predictors) were combined so that each predictor was assigned a relative influence score from 0–100%. A higher relative influence score suggests that a particular predictor is more indicative of an outcome than other predictors. In the context of this study, the effort and reward scale items with the highest relative influence characterize the work stress among this population better than items with lower relative influence. This methodology allowed for novel insights on what factors are contributing the most to work stress across all waste workers sampled, as well as which factors contributed the least.

## 3. Results

### 3.1. Participant Characteristics

A total of 68 workers were recruited. Most participants were male (87%) and between 35 and 54 years of age (40%). Most participants reported having a high school diploma or GED (90%), with 40% reporting they had completed at least some college or higher education or been to trade school. Only 12% of participants were relatively new to the waste industry (<2 years of work experience). There was a roughly uniform distribution of workers by income, although only 16% of workers reported being in a labor union (Table 1).

### 3.2. Scale Reliability

All Cronbach’s alpha coefficients were ≥0.80, the level indicating a satisfying level of internal consistency. The over-commitment scale had the highest Cronbach’s alpha (0.85, 95% CI: 0.75, 0.90), while the effort (0.80, 95% CI: 0.68, 0.89) and reward (0.80, 95% CI: 0.68, 0.86) scales had the same level of internal reliability consistency. Item-total correlation coefficients averaged 0.65 (SD = 0.03), with all coefficients above the threshold of 0.3 defined by Siegrist, Li, and Montano [13]. Coefficients ranged from 0.60 to 0.72, indicating substantial consistency of items defining their respective scales.

### 3.3. Work Stress

Summary responses to the psychometric scales are presented in Table 2. Average effort (7.9 ± 2.4) and reward (19.6 ± 3.8) exceeded the midpoint of each scale (7.5 and 17.5, respectively), with 60% and 72% of workers reporting excessive effort and reward, respectively. However, average over-commitment (14.1 ± 3.8) was lower than the scale midpoint (15), with only 1 in 3 (32%) workers reporting excessive over-commitment. Similarly, average ERI (0.93 ± 0.39) was lower than 1, although values ranged from 0.26 to 2.20. Overall, 37% of workers reported high effort and low reward from work (ERI > 1).

### 3.4. Boosted Regression Trees

Figure 1 displays the relative influence of each effort and reward item on ERI. Constant pressure at work due to heavy workload was the most indicative of ERI among the waste workers (relative influence = 30.2%). Having many interruptions/disturbances while performing the job (24.6%) and the job becoming more and more demanding over the past few years (16.6%) were the next two most indicative factors. A lack of respect from superiors contributed the least to ERI (1%), along with inadequate job promotion prospects (2.6%) and poor job security (2.8%).

### 3.5. Factors of Work Stress

Average responses to the ERI questionnaire by structural work factors such as union status, job site, and income are presented in Table 3. Union workers reported significantly higher effort (9.2 ± 1.7) than non-union workers (7.8 ± 2.4). Conversely, union workers reported significantly lower reward (17.7 ± 4.0) than non-union workers (20.0 ± 3.7).

As such, ERI was significantly higher among union workers (1.22 ± 0.47) than non-union workers (0.88 ± 0.35). Significant differences in effort, reward, and over-commitment were not observed by job site, although significant differences were observed in ERI. Small business waste workers reported the lowest ERI (0.75 ± 0.31), with both county recyclers and landfill workers at an industrial recycling plant reporting average ERIs above 1 (1.03 ± 0.39 and 1.08 ± 0.48, respectively). No significant differences by income were observed among the study population.

### 3.6. Regression Models

Results from bivariate and multivariate models of ERI by the structural factors are presented in Table 4. Among the bivariate models, union workers reported 44% (95% CI: 7%, 92%) higher ERI values than non-union workers. Similarly, landfill workers at the large industrial site reported 44% (0%, 109%) higher ERI values than small business waste. No significant RRs were observed by differences in income.

After adjusting for work experience, education, and sex, union workers still significantly reported 44% (95% CI: 6%, 97%) higher ERI than non-union workers (Model 1). The RR for union increased to 76% (95% CI: 5%, 198%) after further controlling for job site (Model 2). When adjusting for other covariates, landfill workers at an industrial plant no longer reported significantly higher ERI than other job sites. After further adjustment for income, union workers reported 79% (95% CI, 4%, 208%) higher ERI than non-union workers (Model 3).

## 4. Discussion

The purpose of this cross-sectional study was to explore work stress by using the ERI questionnaire among a sample of US solid waste workers in the state of Michigan [13]. To the authors’ knowledge, this is the first published study that has explored ERI as an outcome among formal US solid waste workers—an overlooked but essential worker population. The ERI scale remained reliable among this worker population, with high reliability for effort, reward, and overcommitment as well as high item-total correlation coefficients. Nearly two-thirds of respondents indicated excessive effort at work but over three-quarters reported sufficient reward. Self-reported overcommitment was not notable in this worker population.

With an ERI of nearly one, it appears the typical solid waste worker from our study population felt appropriately rewarded/compensated for their endeavors. However, this does not disprove our first hypothesis that waste workers experience ERI due to great effort but low reward, because nearly 40% of workers had an ERI equal to or greater than one. In a study of ERI among municipal solid waste workers in Taiwan, with a focus on musculoskeletal disorders, the prevalence of upper back and hand/wrist disorders was consistently associated with an ERI equal to or greater than one. Even after adjusting for age, education, marital status, and job title, upper back and hand/wrist disorders were significantly higher for the ERI equal to or greater than one group [24]. This suggests that an ERI greater than or equal to one among waste workers—including our study population—has repercussions beyond perceptions of unfairness, such as musculoskeletal disorders, which are known to be impacted psychosocial factors [7,25], as found by Lin and colleagues in the sample of Taiwanese solid waste workers [24].

From the boosted regression trees, the top three contributors to ERI among this waste worker population were constant pressure, interruptions during work, and work becoming more demanding over the past 3 years. Anecdotally, the owners of the local, small-scale waste disposal site discussed how business has increased during the COVID-19 pandemic. With more people at home generating trash (e.g., food waste, paper waste, household construction waste, cleaning products) and more personal protective equipment (PPE) being used by the general public (e.g., surgical masks, gloves, N95s), the family-owned waste company noted increased work hours, increased workdays, and increased workload. Studies on waste workers in other nations during COVID-19 have also found heavier workloads and greater psychological distress among formal and informal solid waste workers [26,27,28].

Interestingly in the bivariate model, despite the increased workload, waste workers from the small, local, family-owned business reported putting in less effort for the rewards they received compared to their counterparts. Those working at the county recycling facility and industrial, corporate waste site had higher ERI. While no existing studies have explicitly compared the ERI of small business employees to large business employees, and this study lacks the sample size for such a comparison, the literature does support that small companies/enterprises typically have employees with higher self-reported job satisfaction and happiness, even with lower salaries. Determinants of greater satisfaction and happiness among small- and medium-business employees compared to large businesses include direct, supportive relationships with supervisors and colleagues, more independence in shaping their job role, increased fulfillment, and greater recognition [29,30,31].

Union workers reported higher effort with lower reward, even after adjusting for job site, income, and other covariates, compared to their non-unionized counterparts at the large industrial solid waste site. This validates our second hypothesis that unionized workers experience greater ERI. This may be attributed to the historically contentious relationships between unions and employers [32,33,34]. Furthermore, unionized workers tend to be provided with greater education and awareness around labor rights and working conditions; union organizations often advocate for minimum safety and health standards [35,36,37,38,39,40]. An article by Le and colleagues on the association between union membership and perceptions of safety climate among US adult workers across industries found that unionized workers had worse perceptions of their workplace safety climate compared to non-unionized workers [41]. However, the approach to addressing ERI should not be to discourage union membership as there are known benefits of it (e.g., higher wages, safer working conditions) [42]. Unionization movements have gained traction during the COVID-19 pandemic due to inadequate pay and protections relative to working conditions and responsibilities [43,44]. This provides all businesses, including those in solid waste, with an opportunity to assess current reward and compensation systems, and areas for improvement.

When controlling for job sites and other covariates, income resulted in no differences in ERI between the three waste sites, disproving our last hypothesis that workers with higher income would have lower ERI. Even though it was not statistically significant, the trend increased positively for ERI with income. A study by Kêdoté and colleagues found a relationship between increased perceived stress and insufficient income among e-waste workers [11]. However, this study was conducted among informal e-waste workers in West Africa where the workers struggled to meet personal basic and familial needs such as nutrition, housing, and health. As such, this context is not comparable to our sample of US waste workers who earn above poverty level and are afforded protections through the Occupational Safety and Health Administration (OSHA).

Addressing ERI among solid waste workers is not only beneficial for the employees, but also for employers. In other worker populations, ERI was found to have significant relationships with turnover intention, perceived organizational support, overall employee wellbeing, and job satisfaction [45,46,47]. For solid waste employers and occupational health practitioners seeking to mitigate worker stress, organizational-level interventions are needed to address ERI. Examples of organization-based interventions may include the following: administrative-level changes related to workload (e.g., length of pickup routes, length of shifts) to alleviate constant pressure and decrease interruptions during work; consideration of compensation provided to solid waste workers commensurate with their high occupational effort; improving prospects for promotion, higher wages, career advancement, and fulfillment; incorporating activities or policy changes that foster an increased sense of support from supervisors and coworkers; and provision of worker health and wellbeing promotion programs, as well as mental health services [48,49,50].

### Limitations

This study is not without limitations. First, this study was cross-sectional and therefore causation cannot be inferred. Second, from a statistical perspective, the sample size analyzed was relatively small. Third, there is a potential for selection bias as site supervisors had to consent to participate for the study team to sample the workers. Fourth, the study population only encompassed volunteer workers from three solid waste sites—with only one type of each site—within Michigan, which may impact true comparisons between site types, and generalizability to other sectors of the waste industry (e.g., medical waste) and to other geographical areas. Fifth, from the sample, less than a dozen were unionized and all came from the same worksite, which would likely influence ERI.

Since this study had a small sample size with which to determine ERI among waste workers, future research should attempt to recruit a larger sample with multiples of each type of waste site, instead of one of each kind. Additionally, because unionized workers had higher ERI than their non-union counterparts, we recommend further investigating differences in perceptions of ERI through qualitative methods to further elucidate key themes that differentiate these two subpopulations beyond union status. Comparisons of ERI pre- and post-union status could also provide additional insight; union membership may lead to a decreased ERI even if it still exists. Furthermore, a comparison of ERI between small and medium businesses versus large businesses may provide greater insight into how organization size impacts work stress, as well as differences in ERI, between formal waste management sectors (e.g., e-waste, medical waste). This is warranted as it is currently lacking in the literature. Finally, the imbalance between workload and reward may result in poor physical health [15,19]. Thus, we propose a more detailed investigation about the relationship between ERI and physical health outcomes, controlling for lifestyle and pre-existing conditions among formal US solid waste workers.

## 5. Conclusions

Waste workers play a critical role in our society but are often overlooked despite the occupational hazards and demanding working conditions they encounter. Work-related stress among solid waste workers, as measured by ERI, can potentially be mitigated by organizational-level changes, such as increasing reward, alleviating constant pressure, and decreasing work demands. This study shows the need for assessment of psychosocial factors of work among employees in the waste industry.

## Figures and Tables

**Figure 1 ijerph-19-06791-f001:**
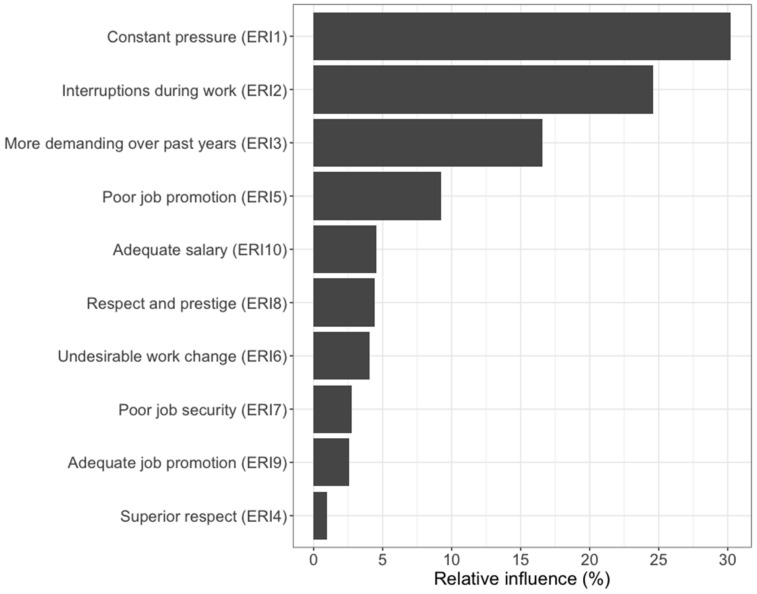
Relative influence of respective effort and reward scale items on effort–reward imbalance.

**Table 1 ijerph-19-06791-t001:** Demographic information.

Scale	N	n (%)
Age	68	
18–34 years		26 (38%)
35–54 years		27 (40%)
55+		15 (22%)
Work experience	68	
<2 years		8 (12%)
2–10 years		44 (65%)
11+ years		16 (23%)
Education	67	
No high school diploma		6 (9%)
High school diploma/GED		34 (50%)
Some college or higher, or trade school		27 (40%)
Sex (male)	68	59 (87%)
Union worker	68	11 (16%)
Job site	68	
Small business waste		9 (13%)
Industrial waste-hauling		36 (53%)
County recycling		8 (12%)
Industrial waste-landfill		15 (22%)
Income	66	
Low (<25 k–<50 k)		23 (34%)
Median (50 k–<75 k)		19 (28%)
High (75 k–>150 k)		24 (35%)

**Table 2 ijerph-19-06791-t002:** Psychometric scale summaries.

Scale	N	Mean (SD)	Min–Max
Effort (scale: 3–12)	68	7.9 (2.4)	3–12
Reward (scale: 7–28)	65	19.6 (3.8)	10–28
Over-commitment (scale: 6–24)	68	14.1 (3.8)	6–24
Effort–reward imbalance (ERI)	65	0.93 (0.39)	0.26–2.20
		**n (%)**	
Excessive effort (>7.5)	68	41 (60%)	
Excessive reward (>17.5)	65	47 (72%)	
Excessive over-commitment (>15)	68	22 (32%)	
High effort, low reward (ERI > 1)	65	24 (37%)	

Note: Higher values of effort, reward, and overcommitment indicate higher degrees of each respective item. ERI = 1 indicates perfect balance between effort and reward; ERI > 1 indicates higher effort, lower reward; ERI < 1 indicates lower effort, higher reward.

**Table 3 ijerph-19-06791-t003:** Psychometric scales by union status, job site, and income.

		Effort	Reward	OC	ERI ^†^
Variable	N Range	Mean (SD)	Mean (SD)	Mean (SD)	Mean (SD)
Union status					
No	55–57	7.8 (2.4)	20.0 (3.7)	13.9 (3.8)	0.88 (0.35)
Yes	10–11	9.2 (1.7)	17.7 (4.0)	15.0 (4.0)	1.22 (0.47)
*p*-value (test for trend)		*p* < 0.05	*p* < 0.05	*p* = 0.39	*p* < 0.05
Job site					
Small business waste	9	7.3 (2.7)	22.1 (3.9)	14.2 (5.0)	0.75 (0.31)
Industrial waste-hauling	34–36	7.7 (2.6)	19.5 (3.5)	13.8 (4.0)	0.89 (0.35)
County recycling	8	8.3 (1.9)	18.6 (3.5)	14.0 (2.5)	1.03 (0.39)
Industrial waste-landfill	14–15	8.6 (2.0)	18.9 (4.1)	14.6 (3.6)	1.08 (0.48)
*p*-value (test for trend)		*p* = 0.14	*p* = 0.09	*p* = 0.66	*p* < 0.05
Income					
Low (<25 k–<50 k)	22–23	7.8 (2.4)	19.6 (3.1)	15.3 (4.2)	0.88 (0.30)
Median (50 k–<75 k)	18–19	7.5 (2.7)	19.8 (5.3)	15.3 (4.2)	0.95 (0.56)
High (75 k–>150 k)	23–24	8.5 (2.2)	19.8 (3.0)	13.6 (3.3)	0.97 (0.32)
*p*-value (test for trend)		*p* = 0.36	*p* = 0.84	*p* = 0.14	*p* = 0.48

Note: OC—over-commitment; ERI—effort–reward imbalance; ^†^ Test for trend on log-transformed values.

**Table 4 ijerph-19-06791-t004:** Relative risk (RR) of structural factors of effort–reward imbalance (ERI) ^†^.

	Bivariate Models	Multivariate Models ^††^		
		Model 1	Model 2	Model 3
Variable	RR (95% CI)	RR (95% CI)	RR (95% CI)	RR (95% CI)
Union worker				
No (ref)	1	1	1	1
Yes	1.44 (1.07, 1.92)	1.44 (1.06, 1.97)	1.76 (1.05, 2.98)	1.79 (1.04, 3.08)
Job site				
Small business waste (ref)	1	--	1	1
Industrial waste-hauling	1.19 (0.86, 1.64)	--	1.16 (0.83, 1.62)	1.18 (0.83, 1.69)
County recycling	1.41 (0.93, 2.15)	--	1.35 (0.86, 2.12)	1.50 (0.92, 2.45)
Industrial waste-landfill	1.44 (1.00, 2.09)	--	0.93 (0.53, 1.62)	0.95 (0.51, 1.76)
Income (US dollars)				
Low (<25 k–<50 k) (ref)	1	--	--	1
Median (50 k–<75 k)	0.98 (0.74, 1.30)	--	--	0.87 (0.64, 1.18)
High (75 k–>150 k)	1.10 (0.84, 1.44)	--	--	0.92 (0.65, 1.33)

^†^ ERI was log-transformed for normality; ^††^ Multivariate models adjusted for work experience, education, and sex.

## Data Availability

The data presented in this study are available on request from the corresponding author. The data are not publicly available due to privacy and confidentiality measures outlined in the approved IRB application.

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
