# Peer review of "Effort–Reward Imbalance among a Sample of Formal US Solid Waste Workers"

_ijerph, 2022, doi:10.3390/ijerph19116791_

Round 1
Reviewer 1 Report
I found this to be an interesting paper.
My comments are as follows:
- The study only covered three sites. This may lead to selection bias as the site supervisors had to consent to participate. How many sites were contacted to get the three sites that agreed? At the individual sites that did participate , what proportion of the workers completed the survey?
- I have a concern regarding the calculation of ERI. It appears that each question is weighed the same given the ratio of 1. I found this confusing as Figure 1 suggests certain questions influence the score more than others. The authors do provide some references for the tool. A stronger explanation of why the tool is valid would be useful. Dividing the scores on effort and reward on the median response and calling imbalance when the effort score is above the median and the reward below may be another way to look at the data.
- In Table 3 the ”N” column is confusing. I believe the lower number is the number of respondents and the higher number the possible responses. Only the lower number is needed.
- The analysis of the union response with higher ERI scores is subject to a number of biases. First there were only 10 union members surveyed. Second these workers appear to coming from one site of which other workers were not union members. Worksite would likely be the major influence on the ERI so it would be interesting to compare these workers to the others who worked in the same establishment. Further, union membership may lead to decrease of ERI even if it still exists . Comparisons of ERI pre and post union affiliation would be useful, but clearly this is beyond the scope of this paper.
Reviewer 2 Report
This study explored the effort-reward imbalance of solid waste workers. It is an interesting topic in the field of work-related stress. There are some comments on improving the study.
Please add detailed information about the variable source in the survey questionnaire.
In the Questionnaire results, the authors should offer more information regarding the sampling. For example, more details on the pilot test should be provided. An adequate sample size should be provided.
